# Nitrogen-Doped Ordered Mesoporous Carbons Supported Co$_3$O$_4$ Composite as a Bifunctional Oxygen Electrode Catalyst

**Jing Wang [1], Shuwei Zhang [1], Haihong Zhong [1,2], Nicolas Alonso-Vante [2], Dianqing Li [1], Pinggui Tang [1] and Yongjun Feng [1,*]**

[1]  State Key Laboratory of Chemical Resource Engineering, Beijing Engineering Center for Hierarchical Catalysts, Beijing University of Chemical Technology, No. 15 Beisanhuan East Road, Beijing 100029, China; 2017200993@buct.edu.cn (J.W.); 2016201069@buct.edu.cn (S.Z.); 2014200922@grad.buct.edu.cn (H.Z.); lidq@mail.buct.edu.cn (D.L.); tangpg@mail.buct.edu.cn (P.T.)

[2]  IC2MP, UMR-CNRS 7285, University of Poitiers, F-86022 Poitiers CEDEX, France; nicolas.alonso.vante@univ-poitiers.fr

*  Correspondence: yjfeng@mail.buct.edu.cn; Tel.: +86-10-6442-5385

**Abstract:** It is increasingly useful to develop bifunctional catalysts for oxygen reduction and oxygen evolution reaction (ORR and OER) for fuel cells, metal-air rechargeable batteries, and unitized regenerative cells. Here, based on the excellent conductivity and stability of ordered mesoporous carbons, and the best ORR and OER activity of Co$_3$O$_4$, the composite Co$_3$O$_4$/N-HNMK-3 was designed and manufactured by means of a solvothermal method, using ordered N-doped mesoporous carbon (N-HNMK-3) as substrate, and then the bifunctional electrocatalytic performance corresponding to ORR, OER in alkaline media was carefully investigated. The results showed that Co$_3$O$_4$/N-HNMK-3 composite, a non-precious metal centered electrocatalyst, displayed excellent ORR performance (activity, selectivity, and stability) close to that of commercial 20 wt.% Pt/C and a promising OER activity near 20 wt.% RuO$_2$/C. The outstanding bifunctional activities of Co$_3$O$_4$/N-HNMK-3 was assessed with the lowest $\triangle$E value of 0.86 V (E$_{OER,10\,mA\,cm^{-2}}$-E$_{ORR,-3\,mA\,cm^{-2}}$) with respect to the two commercial precious metal-based electrocatalysts.

**Keywords:** Ordered mesoporous carbon; nitrogen doping; cobalt-based electrocatalyst; bifunctional oxygen electrode; solvothermal method

## 1. Introduction

Hydrogen is deemed as the best way for renewable sources energy-storage to replace petroleum-based energy resources because of its high calorific value, abundant reserve, and near zero release. Yet, both the oxygen reduction reaction (ORR) and oxygen evolution reaction (OER) individually play crucial roles in fuel cells for the practical application of hydrogen, and water electrolysis for the high-efficiency production of high-purity hydrogen [1–4]. Recently, unitized regenerative cells (URCs), integrating water electrolysis and fuel cell in one setup, have attracted increasing attention due to potential applications in aviation, submarines and transportation as a mobile power source. This issue remains key to exploring high performance and low cost. In terms of electrocatalytic activity, Pt/C and Ru-/Ir oxides are generally considered to be the most advanced electrochemical oxygen catalysts for the ORR and OER processes, respectively. However, high cost, low methanol tolerance, and poor stability continue to hamper practical marketing development [5].

Therefore, it remains a big challenge to develop promising bifunctional oxygen electrodes for the development of URCs.

Among non-precious metal electrocatalysts being developed such as cobalt-based chalcogenides, $CoSe_2$, CoS, and $Co_3O_4$, show outstanding performance toward oxygen electrode reactions including activity, selectivity, and durability in alkaline media [6–9]. However, cobalt-based chalcogenide electrocatalysts have a low electrical conductivity and *per se* a negative effect on the performance of the electrocatalyst. On the other hand, ordered mesoporous carbon as a conductive substrate has attracted more attention because of large surface, high conductivity and long durability. [10–13]. In addition, the doping heteroatom, such as nitrogen [14,15], phosphor [16,17], sulfur [18] and boron [19], can further improve electrocatalytic performance towards oxygen electrode reactions due to new active species. In a previous report, we developed N-doped ordered mesoporous carbons (CMK-3) with high ORR activity, long stability, and outstanding methanol tolerance in 0.1 M KOH [20]. Therefore, it is possible to explore high-performance, low-cost electrocatalysts using N-doped ordered mesoporous carbon as a substrate to support cobalt-based chalcogenides.

Here, we manufactured the N-doped ordered mesoporous carbon substrate (N-HNMK-3) for $Co_3O_4$ composite ($Co_3O_4$/N-HNMK-3) by means of a solvothermal route using the home-made ordered mesoporous carbon as carbon source, melamine as a nitrogen source, cobalt acetate as a cobalt source, and urea as the precipitating agent. Figure 1 describes the corresponding synthesis process. In addition, we carefully examined the bifunctional electrocatalytic performance of ORR and OER processes in alkaline media using the rotating disc electrode technique. The compound, based on non-precious metals, shows a promising electrocatalytic performance.

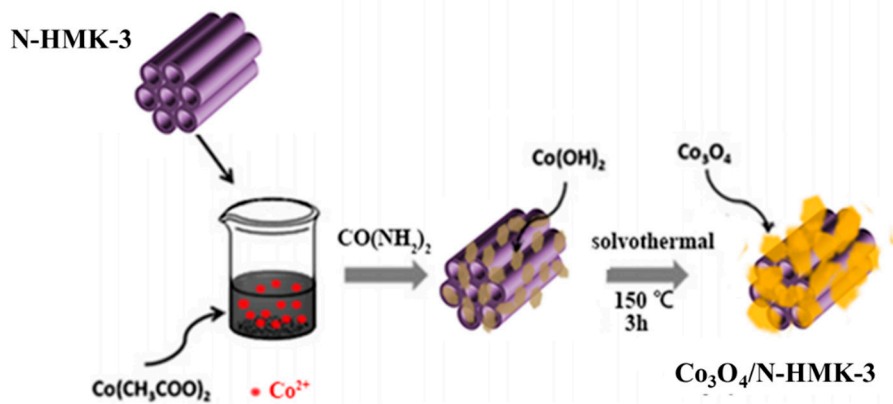

**Figure 1.** Schematic on preparation process of $Co_3O_4$/N-HNMK-3.

## 2. Materials and Methods

### 2.1. Chemicals

Cobalt acetate ($Co(CH_3COO)_2 \cdot 6H_2O$), urea ($CO(NH_2)_2$), $HNO_3$ (6M), melamine ($C_3H_6N_6$), and ethanol ($C_2H_5OH$) were A.R. grade and were directly used as received from Beijing Chemical Co. Ltd (Beijing, China). CMK-3 was synthesized as reported previously [20].

### 2.2. Synthesis of Nitrogen-Doped Ordered Mesoporous Carbon (N-HNMK-3)

Nitrogen-doped ordered mesoporous carbon was fabricated by post-treated nitrogen doping method using ordered mesoporous carbon (CMK-3, surface area, 1247.5 $m^2$/g; pore volume, 1.5 mL/g) as carbon source, and melamine as nitrogen source. To obtain HNMK-3, CMK-3 (0.5002 g) was dispersed in 6.0 M $HNO_3$ (50.0 mL) under vigorous stirring at 70 °C for 3 h, and the resulting suspension was cleaned to neutral pH using deionized water. Thereafter, 0.5000 g HNMK-3 and 2.5000 g melamine were then mixed during grinding to form the homogeneous mixture. Finally, the

obtained mixture was calcined in a nitrogen atmosphere at 900 °C for 4 h to produce nitrogen-doped ordered mesoporous carbon (N-HNMK-3).

## 2.3. Synthesis of $Co_3O_4$/N-HNMK-3 Electrocatalyst

$Co_3O_4$/N-HNMK-3 electrocatalyst was carefully generated via the solvothermal route using N-HNMK-3 as conductive support; $Co(CH_3COO)_2$ as cobalt source, and $CO(NH_2)_2$ as the precipitating agent. Typically, 0.0175 g of the as-prepared N-HNMK-3 was dispersed in ethanol (40.0 mL) by ultrasound for 2h to form the suspension A. Simultaneously, 1.50 g Co(Ac)$_2$ (6.0 mmol) was added in deionized water (40.0 mL) and ultra-sonicated for 20 min to form solution B. Then suspension A and solution B were mixed and ultrasonicated again to produce another suspension. At this point 0.72 g $CO(NH)_2$ was added under vigorous stirring at 80 °C for 24 h to produce a suspension, which was heated at 150 °C in a Teflon-lined stainless-steel autoclave for 3 h. The cooled product was centrifuged and washed with deionized water and ethanol three cycles, and dried at 60 °C overnight. The collected final product was $Co_3O_4$/N-HNMK-3. In addition, $Co_3O_4$, as a reference, was prepared in the same procedure without N-HNMK-3.

## 2.4. Characterization

The materials phase was characterized by Rigaku UItima III XRD (Rigaku Corporation, Tokyo, Japan) with Cu $K_\alpha$ radiation ($\lambda$ = 0.154 nm) at 5 min$^{-1}$ per 2θ. The morphologies were analyzed by Zeiss Supra 55 SEM(Carl Zeiss Jena, Oberkochen, German), a JEOL JEM-2012 TEM (JEOL, Tokyo, Japan), and HRTEM (JEOL, Tokyo, Japan) with a line resolution of 0.19 nm was used. Raman spectra were examined via a Nanofinder 3.0 Raman spectrometer with a He-Ne laser beam of 532 nm. Binding energies of chemical species were collected by XPS using VG ESCALAB 2201 XL spectrometer (Thermo VG Scientific, West Sussex, England).

## 2.5. Electrochemical Measurements

All electrochemical tests were performed in a standard three-electrode system on the Chen Hua electrochemical workstation (CH Instruments Ins., Shanghai, China) at 25 °C. Here, a saturated calomel electrode and a Pt-wire were employed as the reference and counter electrodes, respectively. In this work, all potentials were referred to a reversible hydrogen electrode (RHE): $E_{RHE}$ = $E_{SCE}$ + 0.99 V (0.1 M KOH); and $E_{RHE}$ = $E_{SCE}$ + 1.06 V (1.0 M KOH).

Prior to catalyst deposition, the GC disk was polished with γ-alumina (5A), and successively ultra-sonically treated in water, and then ethanol for one min in each solvent. The ink was prepared by mixing the catalyst powder (6.7 mg), Nafion (5 wt.%) (40 μL), ethanol (300 μL), deionized water (1160 μL) and ultra-sonicated for 40 min. 4.2 μL of the ink was deposited onto the GC disk (4.0 mm, diameter) with a catalyst mass loading of 150 μg cm$^{-2}$. The electrolyte was degassed with argon and oxygen for 30 min before cyclic voltammetry (CV) tests. Also, two benchmarks: 20 wt.% Pt/C (Johnson Matthey) and $RuO_2$/C were prepared as references with a corresponding loading of 60 μg cm$^{-2}$ for Pt, and $RuO_2$.

In the ORR test, the LSV curves were recorded in $O_2$-saturated 0.1 M KOH by scanning the disc potential vs. RHE from 0.95 V to 0.15 V at 5 mVs$^{-1}$ with the electrode rotated at 2500, 2025, 1600,1225, 900, 625, and 400 rpm, and analysis performed using the Koutecky-Levich (K-L) equation:

$$\frac{1}{j} = \frac{1}{j_k} + \frac{1}{j_d} = \frac{1}{j_k} + \frac{1}{B\omega^{\frac{1}{2}}} \qquad (1)$$

where:

$$B = 0.62nFC_0D_0^{2/3}v^{-1/6} \qquad (2)$$

$$j_k = nFkC_0 \qquad (3)$$

$j$, $j_k$, $j_d$ are the corresponding measured, kinetic and diffusion limiting current densities, respectively [5]; $C_0$ and $D_0$ were the saturated concentration, and the diffusion coefficient of $O_2$ ($1.14 \times 10^{-6}$ mol cm$^{-3}$ and $1.73 \times 10^{-5}$ cm s$^{-1}$) in 0.1 M KOH, $v$ was the kinetic viscosity coefficient (0.01 cm$^2$ s$^{-1}$).

In the RRDE test, the $HO_2^-$% and the number of electrons transferred(n) was calculated by the following two equations:

$$HO_2^-\% = 200 * \frac{I_R/N}{I_D + I_R/N} \tag{4}$$

$$n = 4 * \frac{I_D}{I_D + I_R/N} \tag{5}$$

where $I_D$, $I_R$ and $N$ is disk current, ring current and collection efficiency (0.424), respectively.

For the OER tests, measurements were carried out in $O_2$-saturated 1.0 M KOH with a rotating speed of 1600 rpm at a scan rate of 5 mV s$^{-1}$ from 1.0 V to 1.65V *vs.* RHE.

## 3. Results and Discussion

### 3.1. Structure, Morphology, and Chemical Composition

Figure 2 shows the powder PXRD patterns of N-HNMK-3, $Co_3O_4$, and $Co_3O_4$/N-HNMK-3. Compared to CMK-3, the peak of N-HNMK-3 samples at 23.8°/2θ shifts to a lower angle with a higher degree of graphitization, which could be the result of a change in electron distribution following the incorporation of the heteroatom on the surface of carbon materials [21]. Based on the Bragg equation, a series of Bragg reflection peaks in $Co_3O_4$ and $Co_3O_4$/N-HNMK-3 samples correspond to (111), (220), (311), (400), (511), and (440) crystal faces, that match well with the cubic spinel phase $Co_3O_4$ (ICDD PDF card No. 42-1467). This means that the urea-assisted solvothermal method is an available way to prepare the $Co_3O_4$/N-HNMK-3 composite.

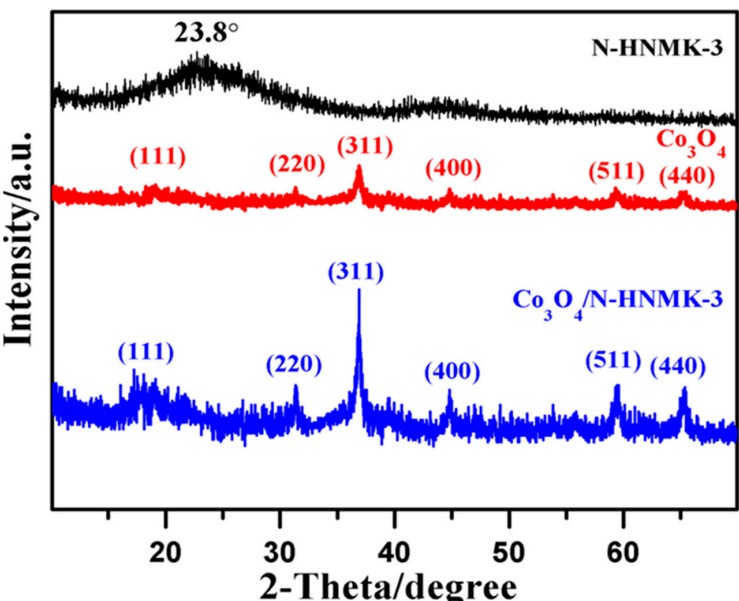

**Figure 2.** Powder XRD patterns of N-HNMK-3, $Co_3O_4$, and $Co_3O_4$/N-HNMK-3 composite.

Figure 3 displays the morphologies and microstructures of $Co_3O_4$ and $Co_3O_4$/N-HNMK-3 samples. The $Co_3O_4$ sample shows different rod-shape in size, Figure 3a. In comparison, the $Co_3O_4$ particles, in $Co_3O_4$/N-HNMK-3 composite, are well-dispersed and supported on N-HNMK-3 substrate, Figure 3b,c. In addition, the average particle size for $Co_3O_4$ was 20–30 nm, Figure 3c. The HRTEM image of $Co_3O_4$ reveals three lattice distances such as 0.286, 0.244 and 0.202 nm, which individually correspond to (220), (311) and (400) $Co_3O_4$ crystal planes, Figure 3d.

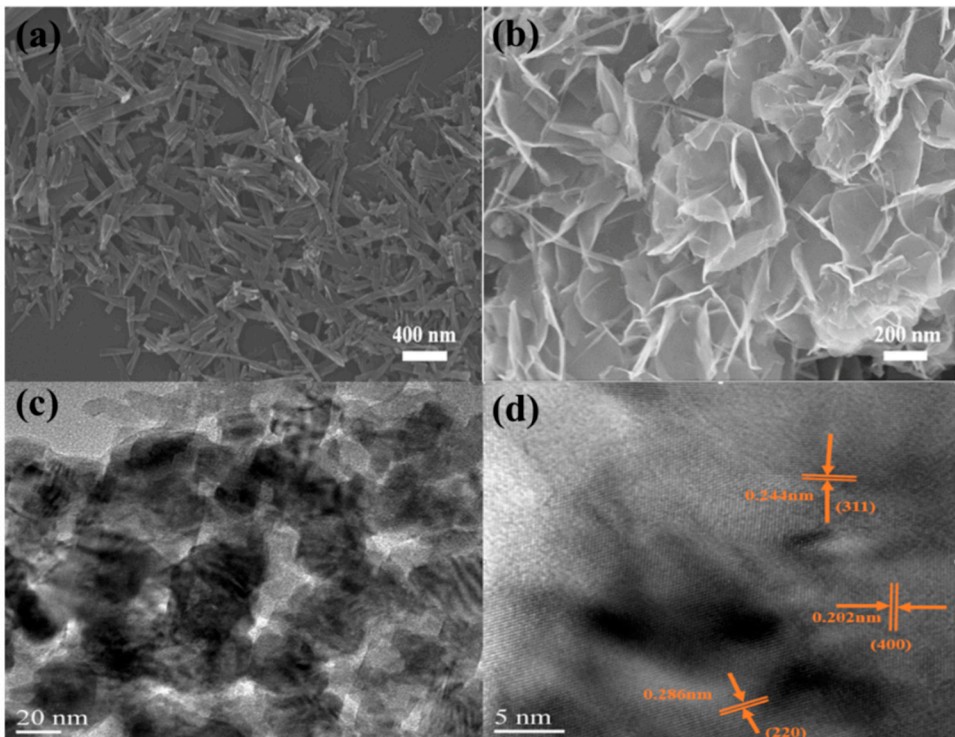

**Figure 3.** (**a**) The SEM images of Co$_3$O$_4$, (**b**) Co$_3$O$_4$/N-HNMK-3. (**c**) TEM and (**d**) HRTEM images of Co$_3$O$_4$/N-HNMK-3.

In addition, Figure 4 shows the Raman spectra of N-HNMK-3, Co$_3$O$_4$, and Co$_3$O$_4$/N-HNMK-3 samples as well as the corresponding fitting profiles of N-HNMK-3 and Co$_3$O$_4$/N-HNMK-3. One observes two characteristic bands for N-HNMK-3, and Co$_3$O$_4$/N-HNMK-3 centered at 1358 (D peak) and 1590 (G peak) cm$^{-1}$. The former peak (D peak) refers to sp$^3$-like C atoms defect sites, whereas the latter (G peak) refers to sp$^2$ C=C stretching mode [22]. Remarkably, other peaks were detected at 192, 468, 516 and 677 cm$^{-1}$, assessing the presence of Co$_3$O$_4$ [23]. The intensity ratio of two bands (I$_D$/I$_G$) illustrates the disordered degree of carbonization. A ratio of 2.96 for N-HNMK-3, and 2.78 for Co$_3$O$_4$/N-HNMK-3 suggest that the incorporation of Co$_3$O$_4$ decreases the disordered degree, increasing the conductivity of the substrate.

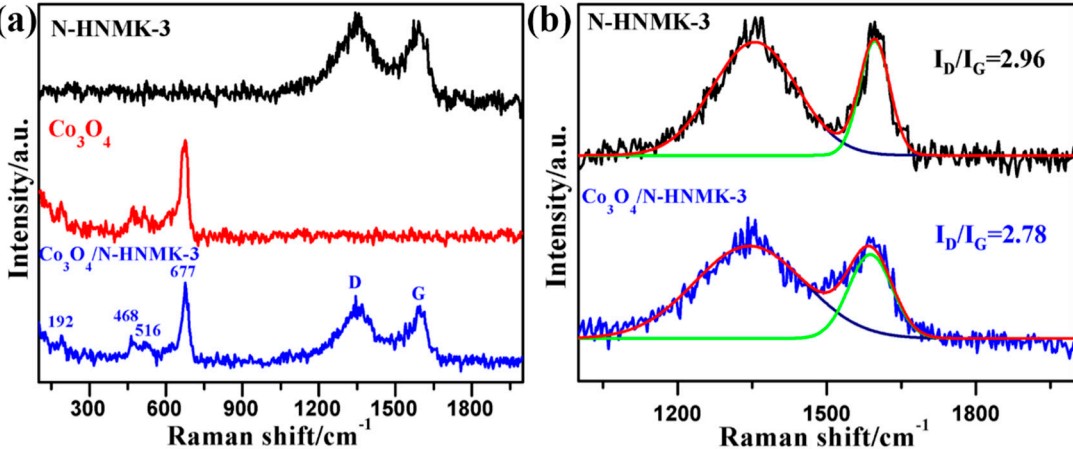

**Figure 4.** (**a**) Raman spectra of N-HNMK-3, Co$_3$O$_4$ and Co$_3$O$_4$/N-HNMK-3. (**b**) The corresponding Raman fitting profiles of N-HNMK-3 and Co$_3$O$_4$/N-HNMK-3.

Surface elemental properties and electronic valence states were further characterized by XPS. A survey spectrum of $Co_3O_4$/HNMK-3 displays the existence of C 1s, N 1s, O 1s, Co 2p signals at 284.5, 399.9, 530.5, 780.1 and 796.2 eV, respectively, as shown in Figure 5a. The corresponding atomic percentage of every element of different samples, N-HNMK-3 has a similar nitrogen content with $Co_3O_4$/HNMK-3, as summarized in Table 1. In the C1s high-resolution XPS spectrum, shown in Figure 5b, the prominent peak centered at 284.6 eV corresponds to $sp^2$-hybridised carbon atoms in graphitic structure. The series of fitted peaks at 285.7, 288.3, and 288.6 eV are attributed to C=N, O-C=O and C-N, respectively. The XPS spectra of C=N and C-N further assess the presence of N-dopant atoms. As to Co2p spectrum, Figure 5c there are two peaks centered at 779.7 and 795.1 eV attributed to Co $2p_{3/2}$ and $2p_{1/2}$ signals, with an energy separation of 15.4 eV, evidencing the presence of $Co_3O_4$ in the composites [24]. Two nitrogen species detected in the N 1s high resolution, shown in Figure 5d, at ~398.6 eV and 400.2 eV are assigned to pyridine and Pyrrole-N. Previous results already demonstrated that pyridine N is an active site for the ORR [25]. The relative content of Pyrrole-N in $Co_3O_4$/HNMK-3 sample is 70.2%, which is higher than that of Pyridine-N (29.8%). It's worth noting that N-HNMK-3 and $Co_3O_4$/HNMK-3 samples have similar N content.

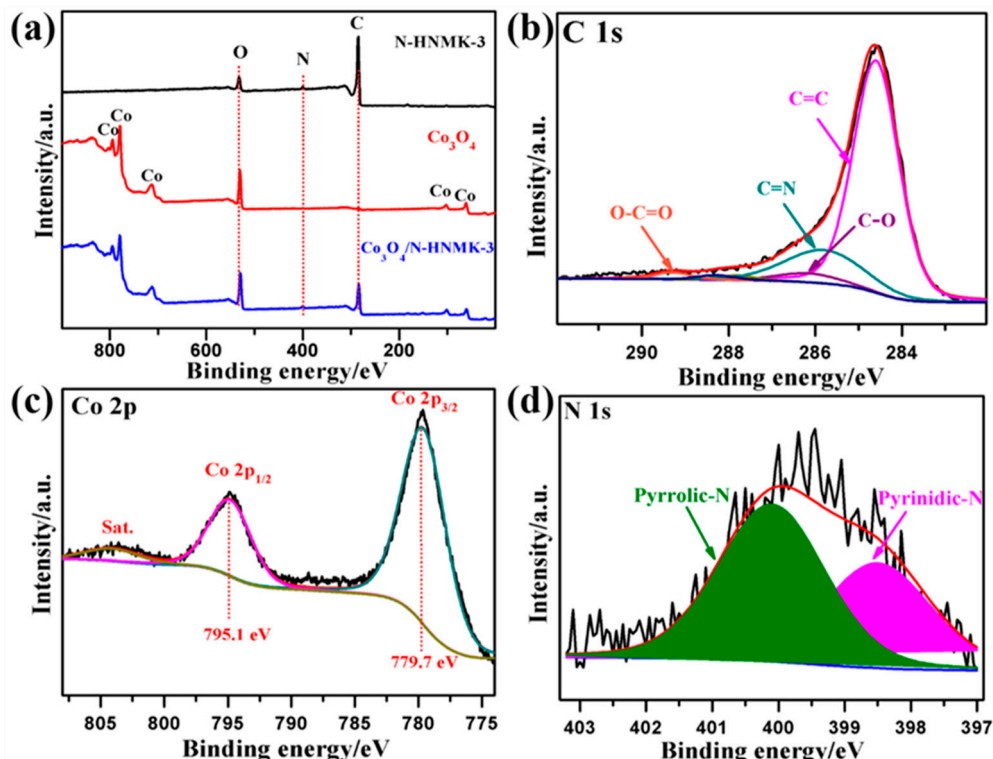

**Figure 5.** (**a**) XPS survey scan of samples: N-HNMK-3, $Co_3O_4$ and $Co_3O_4$/N-HNMK-3; XPS spectra of (**b**) C 1s; (**c**) Co 2p; (**d**) N 1s of $Co_3O_4$/N-HNMK-3.

**Table 1.** The element amount (at. %) in N-HNMK-3, $Co_3O_4$ and $Co_3O_4$/N-HNMK-3 samples.

| Samples | C | N | Co | O | Pyridinic-N | Pyrrolic-N |
|---|---|---|---|---|---|---|
| N-HNMK-3 | 88.8 | 2.6 | | 8.6 | 31.2 | 68.8 |
| $Co_3O_4$ | | | 42.8 | 57.2 | | |
| $Co_3O_4$/N-HNMK-3 | 60.6 | 2.2 | 9.4 | 27.8 | 29.8 | 70.2 |

## 3.2. Electrocatalytic ORR Performance

The ORR performance of $Co_3O_4$/N-HNMK-3 composite was determined by RDE technique, and compared to two commercial reference materials: Pt/C and $RuO_2$/C. Figure 6 displays the activity

towards ORR in 0.1 M KOH electrolyte. Specifically, Figure 6a,b show the ORR activity difference among five samples: N-HMNK-3, physical mixture of N-HMNK-3 and $Co_3O_4$, $Co_3O_4$/N-HMNK-3 composite, Pt/C, and $RuO_2$/C. Herein, it is observed that the $Co_3O_4$/N-HMNK-3 composite has a much higher ORR activity compared to N-HMNK-3, the physical mixture, and $RuO_2$/C, and close to Pt/C. Interestingly, the $Co_3O_4$/N-HNMK-3 composite shows much better ORR activity than the mechanical mixture ($Co_3O_4$ +N-HNMK-3). This difference in the ORR activity results from the possible synergistic effect between two compositions, and from the high dispersion of $Co_3O_4$ on the N-HNMK-3 substrate. In addition, cyclic voltammetry measurement, shown in Figure 6c, was conducted to gain deeper insight into the ORR process. In the $O_2$-saturated electrolyte there is a cathodic peak centered at 0.75 V, which is absent in the Ar-saturated electrolyte. The onset potential of $Co_3O_4$/N-HMNK was detected at 0.90 V *vs.* RHE. The ORR hydrodynamics on $Co_3O_4$/N-HNMK-3 between 400 rpm to 2500 rpm is shown in Figure 6d with well-defined sigmoidal curves. The electron transfers number (n) was estimated through the slopes of fitted K-L plots, see the inset in Figure 6d. Within the applied electrode potential from 0.2 to 0.5 V, the estimated transfer number (n) per $O_2$ molecule was ca. 3.94, suggesting a 4-electron transfer pathway. Meanwhile, we used the RRDE to calculate the hydrogen peroxide and electron transfer number in Figure S1 (See Supporting Information). The low hydrogen peroxide below 15% and transfer number beyond ca. 3.75 in the range of 0.1–0.75 V suggested a close 4-electron transfer pathway.

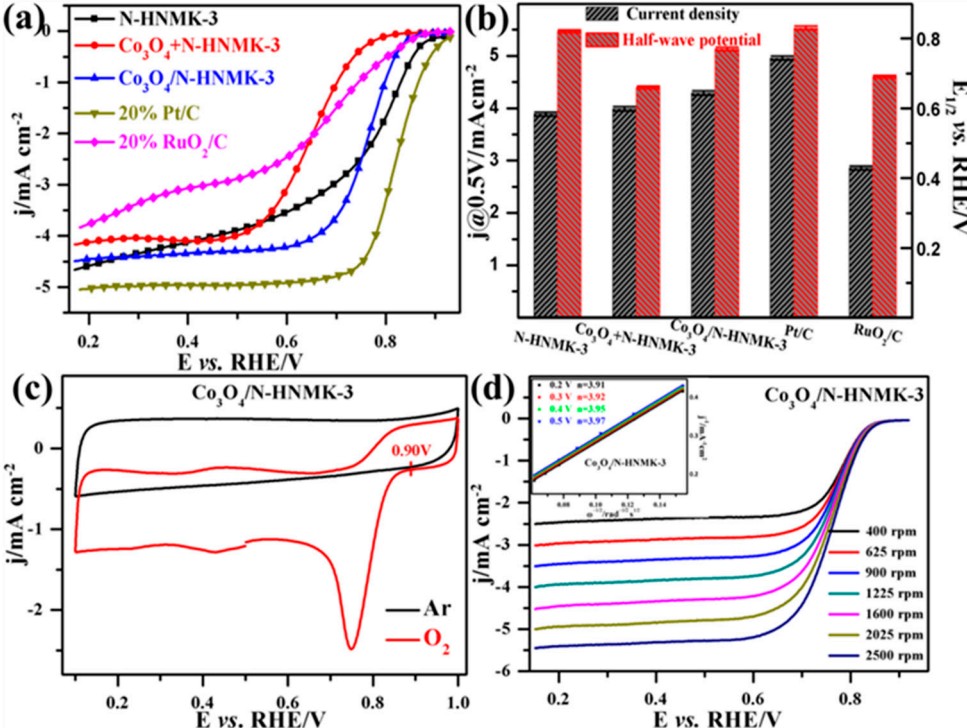

**Figure 6.** (**a**) ORR polarization curves of N-HNMK-3, $Co_3O_4$+N-HNMK-3 (physical mixture), $Co_3O_4$/N-HNMK-3, Pt/C, and $RuO_2$/C at 1600 rpm in $O_2$-saturated 0.1M KOH; (**b**) the limited current density at 0.50 V/RHE and half-wave potential from (**a**); (**c**) CV curves of $Co_3O_4$/N-HNMK-3 in Ar- and $O_2$-saturated 0.1M KOH; (**d**) ORR polarization curves of $Co_3O_4$/N-HNMK-3 at various rotation rates, the inset shows the corresponding K-L plot of $Co_3O_4$/N-HNMK-3.

In addition to activity, the selectivity (tolerance to small organics) and the stability are two other key parameters for a high performance electrocatalyst. Figure 7a depicts the electrocatalytic selectivity in the presence and the absence of 3.0 M methanol in 0.1 M KOH electrolyte. After the addition of 3 M methanol, no significant change was observed in the polarization curves in Ar- and/or $O_2$-saturated

electrolyte, see the inset in Figure 7a. In $O_2$-saturated electrolyte $Co_3O_4$/N-HNMK-3 showed a complete tolerance to methanol. In contrast, as it is well established, Pt/C has no selectivity towards the ORR in presence of methanol, since this material develops a mixed potential due to methanol oxidation and oxygen reduction. Furthermore, the stability test, as demonstrated by chronoamperometry, Figure 7b, was conducted at 0.70 V/RHE, and revealed that, after 18000 s, the ORR activity of $Co_3O_4$/N-HNMK-3 dropped 18.2% against 23.9% on Pt/C. Meanwhile, CVs with multiple cycles in oxygen- saturated solution for ORR were shown in Figure S2 (See Supporting Information), after the accelerated aging test, the reduction potential only shifts 36 mV to a negative region for 1500 cycles.

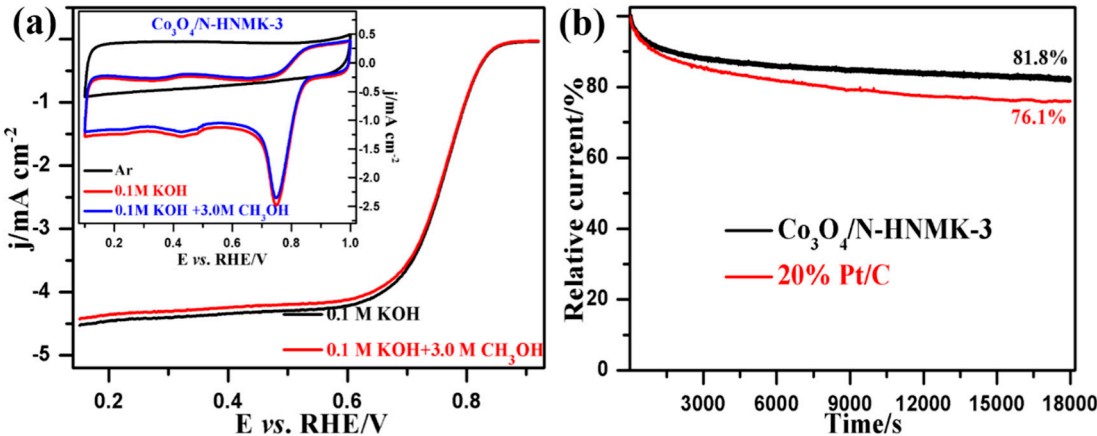

**Figure 7.** (**a**) CV and LSV curves of $Co_3O_4$/N-HNMK-3 in $O_2$- saturated with or without 3.0M methanol in 0.1 M KOH; (**b**) The durability test of $Co_3O_4$/N-HNMK-3 and 20 wt.% Pt/C at 0.70 V in $O_2$-saturated 0.1 M KOH at 1600 rpm.

## 3.3. Electrocatalytic OER Performance

The OER plays a crucial role in water electrolysis and metal-air batteries' systems [26]. In particular, high-performance OER electrocatalysts aim at reducing energy costs to produce high purity hydrogen and promote the development of URCs. Figure 8 shows the activity of $Co_3O_4$/N-HNMK-3 compared to N-HNMK-3, physical mixture of $Co_3O_4$ + N-HNMK-3, Pt/C, and $RuO_2$/C in 1.0 M KOH. Apart from $RuO_2$/C, the $Co_3O_4$/N-HNMK-3 has the highest OER activity with a higher current density of 19.56 mA cm$^{-2}$ at 1.65 V, lower overpotential (365 mV) at the current density of 10 mA cm$^{-2}$, a value close to the best OER electrocatalyst $RuO_2$/C. The Tafel plots for the OER kinetics are shown in Figure 8b. The slope of 93 mV dec$^{-1}$ for $Co_3O_4$/N-HNMK-3 is higher than that of 74 mV dec$^{-1}$ for $RuO_2$/C, and lower than that of the physical mixture. In addition, the OER stability test, via CV, performed on $Co_3O_4$/N-HNMK-3 is shown in Figure 8c,d. No significant variation was observed after 1500 cycles, and to some extent, $Co_3O_4$/N-HNMK-3 showed a ca. 4.1% increase in the overpotential at 10 mA cm$^{-2}$ after 1500 cycles. The results confirm that $Co_3O_4$/N-HNMK-3 is quite durable for the OER in 1.0 M KOH electrolyte.

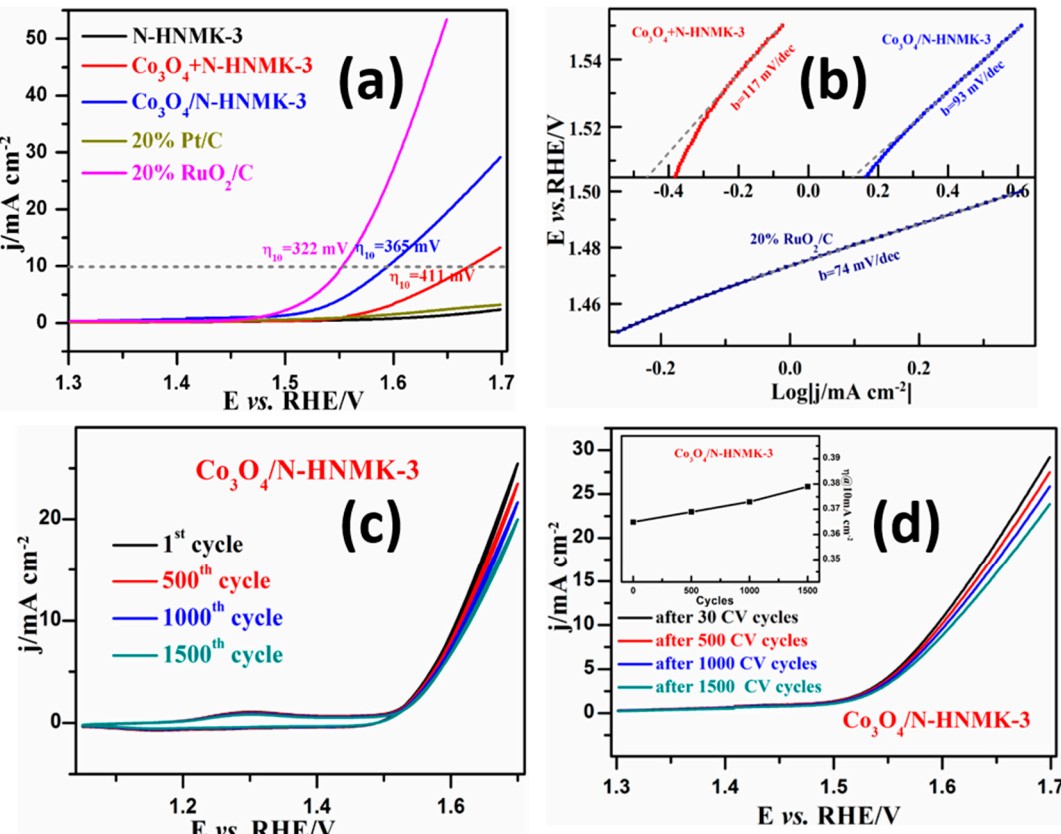

**Figure 8.** (**a**) LSV curves of N-HNMK-3, $Co_3O_4$ + N-HNMK-3 (physical mixture), $Co_3O_4$/N-HNMK-3, 20 wt.% Pt/C, and 20 wt.% $RuO_2$/C towards the OER in $O_2$-saturated 1.0 M KOH at 5 mV s$^{-1}$ and 1600 rpm; (**b**) Tafel plots of each samples derived from curves in (**a**); (**c**) CVs of $Co_3O_4$/N-HNMK-3 from 1.05 V to 1.70 V at 200 mV s$^{-1}$ in $O_2$-saturated 1.0 M KOH from 1st cycle to 1500th cycle; (**d**) LSV curves of $Co_3O_4$/N-HNMK-3 from 1.30 V to 1.70 V at 5 mV s$^{-1}$, the inset presents the variation of $\eta_{10}$ at different CV cycles.

## 3.4. The Bifunctional ORR/OER Performance

The overvoltage between ORR and OER is an important parameter to evaluate the bifunctional electrocatalytic activity of the material. The corresponding data are summarized in Table 2. The ΔE value is defined as the difference in potential between the OER current density of 10.0 mA cm$^{-2}$ and the ORR current density of $-3.0$ mA cm$^{-2}$. Among the listed catalysts [27–32] reported in the literature, $Co_3O_4$/N-HNMK-3 shows the best and promising bifunctional activity with a minimum ΔE value, which is attributed to the interaction developed between $Co_3O_4$ species onto N-doped ordered mesoporous carbons.

**Table 2.** The oxygen bifunctional electrocatalytic performance of catalysts prepared in this work compared with other bifunctional catalysts from the literature.

| Catalysts | E *vs.* RHE at j = −3 mA cm$^{-2}$ | E *vs.* RHE at j = 10 mA cm$^{-2}$ | ΔE ($E_{OER@10\ mA\ cm^{-2}}$)-($E_{ORR@-3\ mA\ cm^{-2}}$) | Refs. |
|---|---|---|---|---|
| N-HNMK-3 | 0.69 | 2.13 | 1.44 | This work |
| $Co_3O_4$+N-HNMK-3 | 0.56 | 1.69 | 1.13 | This work |
| $Co_3O_4$/N-HNMK-3 | 0.73 | 1.59 | **0.86** | This work |
| 20% Pt/C | 0.81 | 2.07 | 1.26 | This work |
| 20% $RuO_2$/C | 0.44 | 1.55 | 1.11 | This work |
| $Co_3O_4$/$Co_2MnO_4$ | 0.68 | 1.77 | 1.09 | 27 |
| $MnCo_2O_4$/Nanocarbon | 0.79 | 1.75 | 0.96 | 28 |
| $NiCo_2O_4$/Graphene | 0.60 | 1.68 | 1.08 | 29 |
| $CoFe_2O_4$/Biocarbon | 0.69 | 1.67 | 0.98 | 30 |
| O-NiCoFe-LDH | 0.62 | 1.67 | 1.05 | 31 |
| $Co(OH)_2CO_3$/C | 0.81 | 1.73 | 0.92 | 32 |

## 4. Conclusions

In this paper, a nitrogen-doped ordered mesoporous carbons supported $Co_3O_4$ composite ($Co_3O_4$/N-HNMK-3) was prepared through a solvothermal method. The $Co_3O_4$/N-HNMK-3 composite showed a much higher ORR performance, which was close to that of the commercial Pt/C in 0.1 M KOH. An excellent OER performance near commercial $RuO_2$/C in 1.0 M KOH was also obtained. Interestingly, $Co_3O_4$/N-HNMK-3 has outstanding bifunctional activity with a much lower △E value between $E_{OER,10\ mA\ cm^{-2}}$-$E_{ORR,-3\ mA\ cm^{-2}}$ as compared to Pt/C and $RuO_2$/C in alkaline medium. Undoubtedly, the synthesis method developed here to obtain the oxygen electrode based on $Co_3O_4$/N-HNMK-3, with non-precious metals will promote the practical development of fuel cells, water electrolysis and unitized regenerative cells.

**Supplementary Materials:** The following are available online at http://www.mdpi.com/2571-9637/2/2/18/s1, Figure S1. (a) Ring (top) and disk (down) current density from RRDE measurements of $Co_3O_4$/N-HNMK-3 samples after annealing at different temperature in $O_2$-saturated 0.1 M KOH at 25 °C with a sweep rate of 5 mV s$^{-1}$ at a rotating speed of 1600 rpm; (b) Molar fraction of $HO_2^-$ formation and electron transfer number n from rotating ring-disk electrode (RRDE) curves in (a). Figure S2. (a) CVs of $Co_3O_4$/N-HNMK-3 from 0.1 V to 1.0 V at 100 mV s$^{-1}$ in $O_2$-saturated 0.1 M KOH from 1st cycle to 1500th cycle; (b) LSVs of $Co_3O_4$/N-HNMK-3 from 0.1 V to 1.0 V at 5 mV s$^{-1}$ in $O_2$-saturated 0.1 M KOH from 1st cycle to 1500th cycle.

**Author Contributions:** Methodology, N.A.-V. and Y.F.; formal analysis, S.Z., G.Y. and H.Z.; investigation, J.W.; writing—original draft preparation, J.W. and S.Z.; writing—review and editing, N.A.-V., D.L., P.T., and Y.F.; supervision, Y.F.; project administration, Y.F.

**Acknowledgments:** This work is supported by the National Natural Science Foundation of China, and the Fundamental Research Funds for the Central Universities (12060093063). H.H. Zhong specially thanks the financial support from China Scholarship Council (201806880040).

**Conflicts of Interest:** The authors declare no conflict of interest.

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
