# Peer review of "Nitrogen-Doped Ordered Mesoporous Carbons Supported Co3O4 Composite as a Bifunctional Oxygen Electrode Catalyst"

_surfaces, doi:10.3390/surfaces2020018_

Round 1
Reviewer 1 Report
This paper describes ORR on several cobalt oxide-nitrogen doped carbon composites. Materials are characterized and their catalytic properties are evaluated. I would like to suggest revising the following:
Please check data in figure 1(b), especially half wave potential. In my opinion, they are not consistent with data in figure 1(a).
Is there any chance to evaluate hydrogen peroxide formation through the positive current around 0.9 V for the different samples? If so, are there any correspondence with the different limiting currents in figure 1(a)?
Authors should explain in detail how real surface areas have been calculated.
Author Response
Please check data in figure 1(b), especially half wave potential. In my opinion, they are not consistent with data in figure 1(a).
AS: The mistake was corrected in the revision.
2. Is there any chance to evaluate hydrogen peroxide formation through the positive current around 0.9 V for the different samples? If so, are there any correspondence with the different limiting currents in figure 1(a)?
AS: Based on the RRDE data in Figure S1(a), the hydrogen peroxide and electron transfer number have been provided in the revision as shown in Figure S1 (b) in the Supporting Information. Here one observes that the limiting current in Fig. 6(d) is the similar with that shown in Figure S1 (a).
3. Authors should explain in detail how real surface areas have been calculated.
AS: In this work, the diameter of the GC disk is 4.0 mm diameter, and the geometry surface area is 0.1256 cm2, which was used to calculate the current density.
Reviewer 2 Report
This manuscript presents a study of novel bifunctional catalysts for ORR/OER with potential use in regenerative unitized fuel cells. The preparation of catalysts and the characterization is carried out well and the suitability of the novel composite catalyst is proven.
Questions/comments
It might be helpful to present CVs with multiple cycles covering the full potential range of ORR/OER in oxygen-saturated solution and in Argon (such as Figure 6c). This might also give insights into the corrosion resistance of the carbon support.
Author Response
This manuscript presents a study of novel bifunctional catalysts for ORR/OER with potential use in regenerative unitized fuel cells. The preparation of catalysts and the characterization are carried out well and the suitability of the novel composite catalyst is proven.
It might be helpful to present CVs with multiple cycles covering the full potential range of ORR/OER in oxygen-saturated solution and in Argon (such as Figure 6c). This might also give insights into the corrosion resistance of the carbon support.
AS: We have updated CVs and LSVs with different cycles of ORR in oxygen-saturated solution in Fig.S2, related to that in Ar-saturated solution. Here we observe a negligible change of 36 mV after 1500 cycles CV scanned in O2-saturated solution. Also, the LSV curves exhibit the corresponding small reduction. The CVs and LSVs of OER has been given in Fig. 8(c) and (d). Therefore, from both of ORR and OER, the catalysts show significantly stable under the investigated conditions.
Round 2
Reviewer 1 Report
It is OK for me.